# Evolution of Interdisciplinary Transition of Care Services in a Primary Care Organization

**DOI:** 10.3390/pharmacy7040164

**Published:** 2019-12-03

**Authors:** William J. Hitch, Irene Park Ulrich, Anne C. Warren, Dow Stick, Danielle Leyonmark, Mackenzie Farrar

**Affiliations:** Department of Family Medicine, Mountain Area Health Education Center, 123 Hendersonville Rd, Asheville, NC 28803, USA; irene.ulrich@mahec.net (I.P.U.); Andy.warren@mahec.net (A.C.W.); dow.stick@mahec.net (D.S.); Danielle.leyonmark@mahec.net (D.L.); mfarrar@iuhealth.org (M.F.)

**Keywords:** transitions, interdisciplinary, pharmacy, care management, nursing, hospital, readmissions

## Abstract

Transitions of care create complex management challenges for providers and leave patients vulnerable to medication errors and hospital readmissions. This article examines the evolution of an interdisciplinary team of pharmacists and nurse care managers and their impact on safe and effective transitions from the acute care settings back into primary care. This article explores successes and challenges of this primary-care-based clinic in managing patients safely through often-complex situations, and explores future directions for improving care processes and outcomes.

## 1. Introduction

Navigating a complex medical system during transitions between acute care and the community can be challenging for both patients and health care providers. Transitions of care involve the movement of a patient from one health care setting to another while providing a continuation of safe and high-quality care [1]. The World Health Organization has identified transitions of care as an important topic in providing safe primary care [2]. Many factors such as lack of appropriate follow-up after discharge, communication deficits, and suboptimal medication therapy during the transitional period can be huge barriers that lead to hospital readmissions [1,3,4]. A recent interview survey by Smeraglio et al. found that readmitted patients and nurse (RN) case managers often felt that factors controlled by the discharge process contributed to their readmission, in 58% and 48% of cases, respectively [5]. Optimizing this process should lead to improved care and decreased readmission rates for patients in these vulnerable positions. 

Many members of the health care team can improve this process. Pharmacists have been able to play a key role in helping to significantly reduce hospital readmission rates and optimize patient outcomes [6,7,8,9], using a variety of different intervention models [10,11]. At hospital discharge, pharmacists are primed to be able to educate patients, intercept medication errors, and resolve issues related to medication access, resulting in improved adherence to treatment plans [12]. For example, a pharmacist-run transition-of-care service at an academic medical center was able to reduce readmissions per month from 27.5 in the year prior to the pharmacist starting to 25 (P = 0.0369) in the year following the pharmacist involvement. The pharmacist was able to educate 1,011 patients and perform 452 interventions [13]. In turn, this reduction in hospital readmission rates can translate to a reduction in health care costs. Nursing care management also has a significant role to play in optimizing transitions. A study by Farrell et al. showed a reduction in hospital readmission rates from 17.9 to 8% through the implementation of a nursing care management phone-based transition of care clinic [14].

The Mountain Area Health Education Center (MAHEC) was established in 1974 to improve training and retention of health care professionals across Western North Carolina. Located in Asheville, MAHEC serves North Carolina’s 16 westernmost counties with seven primary care clinics and includes a family medicine residency program within the largest facility. The MAHEC serves 29,000 patients and utilizes a multi-disciplinary model of care which includes physicians, behavioral health clinicians, pharmacists, nurse care managers (NCM), and dieticians to help optimize care for patients.

The MAHEC also provides inpatient care for its patients who are hospitalized at Mission Hospitals in Asheville, North Carolina. The Family Practice Service (FPS) cares for adults and pediatric patients and provides a full range of inpatient care, seeing patients on the floor unit, intensive care, and labor and delivery. The service is staffed at all times by one attending physician and two medical residents. One faculty pharmacist rounds with the team five days per week.

## 2. Origins

The original idea for a Post-Discharge Clinic (PDC) emerged from a brainstorming session during a continuous quality improvement meeting at MAHEC in 2012. Physicians, nurses, and pharmacists all reported experiences with patients where the medication list in the outpatient electronic record contained inaccuracies. Many participants in the meeting reported a lack of trust in the recorded medication list and frustrations with multiple inaccuracies such as duplicate entries, discontinued medications listed as active, omissions of medications from outside providers, as well as hospital medication changes not accurately reflected in the current list. Providers speculated that medication errors during transitions could negatively affect patient care and even elevate 30-day readmission rates. The group decided to create more structure around patients transitioning from the hospital and the Pharmacy Department was tasked with championing the project. 

The PDC began as a pharmacy residency project, with a plan to assess baseline rates of 30-day readmissions to the hospital before and after the implementation. Initially, PDC was offered two half-days per week. The creation and early evaluation of this service has been published previously [15]. Briefly, the intervention had three components. 

Component one consisted of patient identification and nurse care manager contact. The MAHEC patients who were hospitalized at Mission Hospital and cared for on the FPS were potentially eligible for PDC. The medical resident would identify patients that were eligible if they were on five or more medications and discharging to home and not to a skilled facility. The resident would send a message to a NCM who would then call the patient at home within two business days of discharge from the hospital. The NCM assessed patient reported health status, and arranged pharmacy and physician follow up. 

Component two consisted of medication reconciliation by the pharmacist. The pharmacist would call the patient or caregiver to discuss the current health status of the patient and assess their understanding of what occurred during the hospital stay. The pharmacist assessed the accuracy of the patient’s current medication list based on hospital records, outpatient records, and fill history at the pharmacy. The pharmacist discussed the results of studies performed in the hospital, counseled on any new medications, identified and resolved any medication related problems, assessed and resolved any medication access and adherence issues. Physician follow up was confirmed prior to ending the call. Documentation of this phone call was recorded as an electronic health record (EHR). 

Component three involved prompt physician follow-up after hospitalization. Patients were identified as moderate or high complexity, which dictated the timing of their office visit. The MAHEC defined moderate complexity as having 3–5 active diagnoses in the hospital or 5–9 medications. High-risk patients were defined as those with six or more active diagnoses, 10 or more medications, or an admitting diagnosis of dyspnea. Moderate-risk patients were seen for follow up within 14 days of discharge. High-risk patients were scheduled for follow up within seven days. Patients were preferentially scheduled with their primary care provider or another provider who was familiar with their hospital course, when possible. 

## 3. Evolution

While the PDC pilot model was successful, there remained opportunities to improve utilization of the service. The first identified area for improvement was streamlining the process of reaching the patient. In the pilot, a NCM would call patients and schedule them for a post-discharge phone call at a later date. While in theory this call would give the patient advanced notice, allowing them to be available and prepared with their medications, in practice the extra step did not seem to improve the rate at which patients answered PDC phone calls. Patients who did answer calls were not always prepared for the call with their medications accessible. To address this issue, PDC was adapted in October 2017 to remove the NCM screening phone call and expanded to be offered three half-days per week, Mondays, Wednesdays, and Fridays, rather than just on two days. By offering PDC every other business day, all new patient discharges were being called within 48 h. One concern with removing the NCM screening component was that many patients would be unavailable for calls. However, the authors’ experience proved otherwise, with a similar proportion of patients answering calls compared to when these calls were formally scheduled. 

Removing the screening call and increasing the availability of PDC also enabled expansion of the service to other payer types. The pilot period focused on patients insured by Medicare, which allowed for billing of transitional care management (TCM) codes for this service. Following a successful pilot period, it became clear that this service should be provided to all patients, regardless of payer type. This expansion meant an increase in the volume of calls in PDC, so the increase from two to three days per week was necessary. 

Additionally, modifications were made to how patients requiring PDC services were identified. Initially, medical residents were responsible for identifying patients who met the criteria. This process was inconsistently followed and relied on extremely busy residents to complete additional tasks. It also only captured MAHEC patients on FPS and not on other hospital service lines such as orthopedics, cardiology, or neurology. To address this, a daily report of MAHEC patients discharged from Mission Hospitals was created. A scheduler would run this report, and identify which patients met criteria to receive a PDC phone call. 

While the expansion of PDC indicated larger clinical reach of the program, the growth in call volume often pulled pharmacists from other clinical activities including direct patient care, precepting, and population health work. It became clear that additional help to meet the needs of patients in transition was needed. 

Concurrent with the expansion of PDC, our practice was also rebuilding a care management team led by nursing, which had been lost since the inception of the PDC project. Recognizing that care management concerns, including access to medications, were issues often encountered during PDC, the care management team was engaged in the transitions of care process as well. Initially, the NCM participated by making post-discharge phone calls one to two half-days per week. Because of the success with care management and expansion of their team, PDC was restructured to be primarily managed by the NCM in April 2019, with a fulltime position devoted to PDC. In addition, pharmacy continues to provide assistance two half-days per week, utilizing students and resident learners who assist in making post-discharge calls. Additionally, a referral process for specific patients with significant medical complexity was created to have pharmacists address medication-related problems. The NCM could suggest further outreach via phone or help coordinate having a pharmacist see the patient in person as part of the physician visit. This team-based model allowed for a focus on the respective skill sets of each discipline and provided more comprehensive care for our patients in transition.

## 4. Successes

The PDC is a model that is continuing to grow and change, and through these changes there have been many initiatives that have proved successful. With the initiation of PDC in 2012, an assessment of readmission rates both at baseline and after implementation of the program was conducted, which has been published previously [15]. This pilot assessed a random sample of patients hospitalized prior to the existence of PDC, and assessed whether they were readmitted within 30 days. This sample was compared to another random sample of patients admitted to the hospital one year later, after the introduction of PDC. The baseline sample had a readmission rate of 14.2%. One year later the readmission rate of the second sample was 5.3%. This research indicated that interdisciplinary efforts to improve transitional care were impacting readmission rates significantly. 

As the oversight of PDC was transitioned to a NCM team, several tools were developed in order to facilitate this process and create consistent workflow. One document created a formalized internal process for conducting the phone call with patients, billing, and documentation in the EHR. This instruction guide helped to ensure consistency in expectations as new members joined the team and other pharmacists worked in PDC. Also, pharmacy students who participated in PDC phone calls while on their advanced pharmacy practice rotations utilized the document to understand the steps to take before and after a phone encounter with a patient. This tool allowed the student to more seamlessly integrate into the process of making post-discharge phone calls and provided clear expectations for items to discuss with the patient, improving consistency. A second document streamlined the process for referrals to a pharmacist for medically complex patients experiencing transitions of care. This protocol allowed for the care management team to independently manage the patients in PDC, but also to seek assistance for particular patients needing further medication education, disease state management, and medication access. 

Due to careful, periodic reevaluation of PDC since its inception, the breadth of service has been significantly expanded. While initial efforts targeted a small population of Medicare patients on FPS, now all patients with Medicaid, private insurance, and the uninsured are called. Additionally, patients on other service lines are captured, including orthopedics, cardiology, general surgery and others. Coordination with the regional accountable care organization is occurring to utilize reports that help identify patients discharged from hospitals other than Mission, as well as to coordinate with some skilled nursing facilities to identify when patients are transitioning back into primary care. These changes in process as well as the addition of care management have significantly increased the volume of patients receiving PDC services during transitions of care. 

## 5. Challenges

Despite many positive steps in the right direction, there are several challenges that continue complicate the care of patients during transitions. The first and largest hurdle is information sharing. Many organizations involved in the transition process including hospitals, primary care practices, skilled nursing facilities, and pharmacies have pertinent information within their own siloed EHRs. The practice currently has access to Mission Hospital’s EHR and that of two partnered skilled nursing facilities, but other institutions require more proactive communication and medical records requests. Due to these barriers, it is often not known when patients are discharged from hospitals or skilled nursing facilities, making current medication lists difficult to access, and reasons for certain medication changes unclear. This can significantly hamper the ability of the team to see them in a timely fashion and provide high-quality care. 

Another challenge involves reaching our patients by phone. Despite numerous efforts, some patients never answer the PDC phone call, and medication reconciliation is delayed until their follow-up appointment. Having the wrong number in the medical record and interruptions in patients’ phone services impact the number of patients reached in a phone-based clinic. Also, complex instructions such as insulin administration and inhaler device technique are often better explained face-to-face than over the phone. These delays and challenges can sometimes negatively impact care until patients can be seen in person.

Other operational challenges within the primary care practice involve standardizing processes for organizing the record requests especially from skilled nursing facilities, prioritizing the calls based on fluctuating acuity and volume, staffing of health care team members able to address the growing volume of calls, and standard cross-training of team members including NCM, pharmacists and professional students. 

## 6. Future Directions

As PDC evolves, three particular areas of importance are prioritization of patients most likely to benefit from PDC, coordination of care between disciplines and agencies, and attention to social determinants of health. 

Since staffing limitations do not allow outreach to every patient undergoing a care transition, work is required to refine ways to identify which patients are most in need of PDC phone calls. Much of this is based on clinical judgement in reviewing the patient’s hospital discharge summary and clinic medical record, looking for uncontrolled, high-risk diagnoses such as chronic obstructive pulmonary disease, heart failure, diabetes, and sepsis; as well as other factors such as comorbidities, polypharmacy, and frequent hospitalizations. A major challenge in identifying and prioritizing patients that is present across the health care system is the lack of a centralized medical record system. Better integration of EHRs across health care settings and agencies would drastically improve care transitions. 

Another factor to consider in prioritization is the availability of other NCM services for certain populations; for example, a patient who has Medicaid, certain Medicare payers, or is pregnant may qualify and be referred for TCM by other agencies in the local area. Ongoing communication and relationship-building with these agencies is increasingly important. 

With the involvement of nursing in transitions of care, there is also an increased focus on a broader range of patient needs in the transitional care period such as access to durable medical equipment (DME), referrals to specialists, caregiver involvement, resource linkage, and other social needs. Frequently it falls to the NCM to ensure these linkages are made and coordinate with a variety of service providers, both within the clinic (i.e., physicians, pharmacists, schedulers, dieticians, and mental health providers) and outside the clinic (i.e., home health nurses, DME providers, retail pharmacies, skilled nursing facilities, and specialists).

With the implementation of managed Medicaid in North Carolina over the next year, it is expected that there will be many further changes in how transitional care is provided in the local area, the relationships of agencies involved, and the resources available to the most vulnerable patients, all of which will necessitate ongoing adaptation of our approach. The PDC will continue to grow and evolve to meet the needs of the complex and diverse patient population in western North Carolina.

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
