# Peer review of "Evolution of Interdisciplinary Transition of Care Services in a Primary Care Organization"

_pharmacy, 2019, doi:10.3390/pharmacy7040164_

Round 1

Reviewer 1 Report

Thank you for the opportunity to contribute to the peer review process for the 'Opinion Paper or Communication' submission entitled ‘Evolution of Interdisciplinary Transition of Care Services in a Primary Care Organization’ (pharmacy-643077) from Hitch et al. The authors have presented their experiences around introducing an interdisciplinary primary care clinic to facilitate transitional care from the acute care setting.

On the whole the submission is well constructed and presented. The following are offered as suggestions to further enhance the dissemination of their message to the journal audience:

Page 2 of 6, Introduction, line 48 - suggest amending to avoid starting sentence with an abbreviation 'The MAHEC serves...'

line 52- suggest amending to avoid starting sentence with an abbreviation 'The MAHEC also...'

line 54 - suggest review phrase 'on the floor' for clarity as the terminology 'floor unit' may not be familiar / understood by all readership of the journal.

line 55 - suggest amending to '...delivery wards.' Further, does 'at all times' mean 24-hours, 7-days? The reviewer asks as it unusual to have access to an on-site pharmacist 24-7 even in ICU & ED - often this is on-call, off-site support in many jurisdictions.

Origins, line 58 - suggest supplement to '... for a Post...

line 59 - suggest no point in defining abbreviation here as it is not used again within submission.

line 67 - suggest capitalize 'Pharmacy Department'

line 68 - suggest amending to avoid starting sentence with an abbreviation 'The PDC began...'

line 72 - suggest amending to avoid starting sentence with an abbreviation 'The MAHEC patients...'

line 73 - suggest supplementing to '...on the FPS...'

line 88 - suggest amending to avoid starting sentence with an abbreviation 'The MAHEC defined...'

line 93 - the abbreviation 'PCP' is not defined prior to use here, but also then is not used again regardless. Therefore it is suggested that this be written in full here.

Page 3 of 6, Evolution, line 107 - suggest amending to depersonalize 'However, the authors' experience...'

line 110 - suggest amending to utilize previously defined abbreviation '... of PDC also...'

line 116 - suggest amending to depersonalize 'Additionally, modifications were made to how patients requiring PDC services were identified.'

line 125 - suggest amending to depersonalize 'It became clear that additional help to meet the needs of patients in transition was needed.'

line 127 - suggest amending to utilize previously defined abbreviation and to depersonalize '...expansion of the PDC was also a rebuilding of practice incorporating a care...'

line 132 - suggest amending to utilize previously defined abbreviation '...team, PDC was...'

lines 137 -8 - suggest amending to depersonalize '...allowed for a focus on...care for patients in transition.'

Success, line 140 - suggest amending to avoid starting sentence with an abbreviation 'The PDC is...'

lines 141-2 suggest amending to depersonalize '...2012, an assessment of readmission... program was conducted, which has...'

Page 4 of 6, line 146 -  suggest amending to depersonalize 'The baseline...'

line 147 - suggest amending to depersonalize '...that interdisciplinary...'

line 162 - suggest amending to depersonalize '...inception, the breadth of the service has been significantly expanded.'

lines 163-4 - suggest amending to depersonalize 'While initial efforts...FPS, now all patients with...uninsured are called.'

line 165 - suggest amending to depersonalize 'Additionally, patients on service lines are captured, including...'

line 166 - suggest amending to depersonalize 'Coordination with accountable care is occurring to utilize...'

Challenges, lines 176-7 - suggest amending to depersonalize 'The practice currently has...two partnered skilled nursing facilities, but...'

line 178 - suggest amending to depersonalize '...barriers, often it is not known when patients...'

line 189 - suggest amending to depersonalize '...within the primary...'

Page 5 of 6, Future Directions, line 199 - suggest amending to depersonalize '...transition, work is required to refine...'

line 207 - suggest amending to depersonalize '...in the local area.'

line 219 - suggest amending to depersonalize '...in the local area, the...'

line 221 - suggest amending to avoid starting sentence with an abbreviation 'The PDC will...'

How about a recommendation around integration or centralization of the EHR (rather than each organization having their own version), or at least improved access between organizations to the various EHRs that exist for clients?

References - accepting impossibility to include all available references in a submission such as this, it may have reasonably been expected to see reference to articles such as (but not limited to) the following:

Transitions of Care: Technical Series on Safer Primary Care. Geneva: World Health Organization; 2016. Licence: CC BY-NC-SA 3.0 IGO. (available at https://apps.who.int/iris/bitstream/handle/10665/252272/9789241511599-eng.pdf;jsessionid=5A14534908686E1086EDDBA8D8645F64?sequence=1)

Leppin, A., Gionfriddo, M., Kessler, M., Brito, J., Mair, F., Gallacher, K., & Montori, V. (2014). Preventing 30-day hospital readmissions: A systematic review and meta-analysis of randomized trials. JAMA Internal Medicine, 174(7), 10951107. doi:10.1001/jamainternmed.2014.1608

Thomas R, Huntley AL, Mann M, et al. Pharmacist-led interventions to reduce unplanned admissions for older people: a systematic review and meta-analysis of randomised controlled trials. Age Ageing 2014;43:17487.doi:10.1093/ageing/aft169

Rodrigues, C., Harrington, A., Murdock, N., Borzadek, E., Calabro, K., Martin, J., & Slack, M. (2017). Effect of pharmacy-supported transition-of-care interventions on 30-day readmissions: A systematic review and meta-analysis. Annals of Pharmacotherapy, 51, 886889. doi:10.1177/1060028017712725

Yu, J., Pincus, K., & Mattingly, T. J., 2nd. (2017). Service description and analysis for an interprofessional discharge clinic within a primary care practice. Journal of Interprofessional Care, 31(6), 771773. doi:10.1080/13561820.2017.1347611 

Mekonnen AB, McLachlan AJ, Brien JA. Effectiveness of pharmacist-led medication reconciliation programmes on clinical outcomes at hospital transitions: a systematic review and meta-analysis. BMJ Open 2016;6:e010003.doi:10.1136/bmjopen-2015-010003

Renaudin P, Boyer L, Esteve MA, et al. Do pharmacist-led medication reviews in hospitals help reduce hospital readmissions? A systematic review and meta-analysis. Br J Clin Pharmacol 2016;82:166073.doi:10.1111/bcp.13085

Joseph T, Hale GM, Eltaki SM, et al. Integration Strategies of Pharmacists in Primary Care-Based Accountable Care Organizations: A Report from the Accountable Care Organization Research Network, Services, and Education. Journal Of Managed Care & Specialty Pharmacy. 2017;23(5):541-548. doi:10.18553/jmcp.2017.23.5.541.

Shelley Otsuka, Jennifer N. Smith, Laura Pontiggia, Radha V. Patel, Susan C. Day & David T. Grande (2019) Impact of an interprofessional transition of care service on 30-day hospital reutilizations, Journal of Interprofessional Care, 33:1, 32-37, DOI: 10.1080/13561820.2018.1513466

Author Response

The author should describe exactly the period in which the study was conducted, as well as details regarding the inclusion and exclusion criteria applied to this study.

We have added dates to better describe the timeline of the evolution of our PDC service. As this was a reflection of our experience, inclusion and exclusion criteria are not applicable.

Reviewer 2 Report

Dear author, the manuscript touches a interesting topic.

It was a pleasure to read such a well written manuscript.

The author should describe exactly the period in which the study was conducted, as well as details regarding the inclusion and exclusion criteria applied to this study.

Best regards!

Author Response

Thank you for the opportunity to contribute to the peer review process for the 'Opinion Paper or Communication' submission entitled ‘Evolution of Interdisciplinary Transition of Care Services in a Primary Care Organization’ (pharmacy-643077) from Hitch et al. The authors have presented their experiences around introducing an interdisciplinary primary care clinic to facilitate transitional care from the acute care setting.

On the whole the submission is well constructed and presented. The following are offered as suggestions to further enhance the dissemination of their message to the journal audience:

Page 2 of 6, Introduction, line 48 - suggest amending to avoid starting sentence with an abbreviation 'The MAHEC serves...'

This edit has been made.

line 52- suggest amending to avoid starting sentence with an abbreviation 'The MAHEC also...'

This edit has been made.

line 54 - suggest review phrase 'on the floor' for clarity as the terminology 'floor unit' may not be familiar / understood by all readership of the journal.

This edit has been made.

line 55 - suggest amending to '...delivery wards.' Further, does 'at all times' mean 24-hours, 7-days? The reviewer asks as it unusual to have access to an on-site pharmacist 24-7 even in ICU & ED - often this is on-call, off-site support in many jurisdictions.

This has been edited to clarify that clinical pharmacists round with the team 5 days per week.

Origins, line 58 - suggest supplement to '... for a Post...

This edit has been made.

line 59 - suggest no point in defining abbreviation here as it is not used again within submission.

This edit has been made.

line 67 - suggest capitalize 'Pharmacy Department'

This edit has been made.

line 68 - suggest amending to avoid starting sentence with an abbreviation 'The PDC began...'

This edit has been made.

line 72 - suggest amending to avoid starting sentence with an abbreviation 'The MAHEC patients...'

This edit has been made.

line 73 - suggest supplementing to '...on the FPS...'

This edit has been made.

line 88 - suggest amending to avoid starting sentence with an abbreviation 'The MAHEC defined...'

This edit has been made.

line 93 - the abbreviation 'PCP' is not defined prior to use here, but also then is not used again regardless. Therefore it is suggested that this be written in full here.

This edit has been made.

Page 3 of 6, Evolution, line 107 - suggest amending to depersonalize 'However, the authors' experience...'

This edit has been made.

line 110 - suggest amending to utilize previously defined abbreviation '... of PDC also...'

This edit has been made.

line 116 - suggest amending to depersonalize 'Additionally, modifications were made to how patients requiring PDC services were identified.'

This edit has been made.

line 125 - suggest amending to depersonalize 'It became clear that additional help to meet the needs of patients in transition was needed.'

This edit has been made.

line 127 - suggest amending to utilize previously defined abbreviation and to depersonalize '...expansion of the PDC was also a rebuilding of practice incorporating a care...'

This edit has been made.

line 132 - suggest amending to utilize previously defined abbreviation '...team, PDC was...'

This edit has been made.

lines 137 -8 - suggest amending to depersonalize '...allowed for a focus on...care for patients in transition.'

This edit has been made.

Success, line 140 - suggest amending to avoid starting sentence with an abbreviation 'The PDC is...'

This edit has been made.

lines 141-2 suggest amending to depersonalize '...2012, an assessment of readmission... program was conducted, which has...'

This edit has been made.

Page 4 of 6, line 146 -  suggest amending to depersonalize 'The baseline...'

This edit has been made.

line 147 - suggest amending to depersonalize '...that interdisciplinary...'

This edit has been made.

line 162 - suggest amending to depersonalize '...inception, the breadth of the service has been significantly expanded.'

This edit has been made.

lines 163-4 - suggest amending to depersonalize 'While initial efforts...FPS, now all patients with...uninsured are called.'

This edit has been made.

line 165 - suggest amending to depersonalize 'Additionally, patients on service lines are captured, including...'

This edit has been made.

line 166 - suggest amending to depersonalize 'Coordination with accountable care is occurring to utilize...'

This edit has been made.

Challenges, lines 176-7 - suggest amending to depersonalize 'The practice currently has...two partnered skilled nursing facilities, but...'

This edit has been made.

line 178 - suggest amending to depersonalize '...barriers, often it is not known when patients...'

This edit has been made.

line 189 - suggest amending to depersonalize '...within the primary...'

This edit has been made.

Page 5 of 6, Future Directions, line 199 - suggest amending to depersonalize '...transition, work is required to refine...'

This edit has been made.

line 207 - suggest amending to depersonalize '...in the local area.'

This edit has been made.

line 219 - suggest amending to depersonalize '...in the local area, the...'

This edit has been made.

line 221 - suggest amending to avoid starting sentence with an abbreviation 'The PDC will...'

This edit has been made.

How about a recommendation around integration or centralization of the EHR (rather than each organization having their own version), or at least improved access between organizations to the various EHRs that exist for clients?

This is an excellent point. We have added language highlighting this challenge throughout the health care system.

References - accepting impossibility to include all available references in a submission such as this, it may have reasonably been expected to see reference to articles such as (but not limited to) the following:

Thank you for these reference. We have commented on them below. We have also added references by Hawes, et al and Rafferty et al.

Transitions of Care: Technical Series on Safer Primary Care. Geneva: World Health Organization; 2016. Licence: CC BY-NC-SA 3.0 IGO. (available at https://apps.who.int/iris/bitstream/handle/10665/252272/9789241511599-eng.pdf;jsessionid=5A14534908686E1086EDDBA8D8645F64?sequence=1) Cited

Leppin, A., Gionfriddo, M., Kessler, M., Brito, J., Mair, F., Gallacher, K., & Montori, V. (2014). Preventing 30-day hospital readmissions: A systematic review and meta-analysis of randomized trials. JAMA Internal Medicine, 174(7), 1095–1107. doi:10.1001/jamainternmed.2014.1608 Cited

Thomas R, Huntley AL, Mann M, et al. Pharmacist-led interventions to reduce unplanned admissions for older people: a systematic review and meta-analysis of randomised controlled trials. Age Ageing 2014;43:174–87.doi:10.1093/ageing/aft169

As this reference referred specifically to older adults and the role of the inpatient pharmacists with hetergenous findings, we opted not to cite this article.

Rodrigues, C., Harrington, A., Murdock, N., Borzadek, E., Calabro, K., Martin, J., & Slack, M. (2017). Effect of pharmacy-supported transition-of-care interventions on 30-day readmissions: A systematic review and meta-analysis. Annals of Pharmacotherapy, 51, 886–889. doi:10.1177/1060028017712725 Cited

Yu, J., Pincus, K., & Mattingly, T. J., 2nd. (2017). Service description and analysis for an interprofessional discharge clinic within a primary care practice. Journal of Interprofessional Care, 31(6), 771–773. doi:10.1080/13561820.2017.1347611 

Because the sample size of the intervention group in this study was so small (13 vs. 154 in the usual care group), we opted to exclude this study.

Mekonnen AB, McLachlan AJ, Brien JA. Effectiveness of pharmacist-led medication reconciliation programmes on clinical outcomes at hospital transitions: a systematic review and meta-analysis. BMJ Open 2016;6:e010003.doi:10.1136/bmjopen-2015-010003 Cited

Renaudin P, Boyer L, Esteve MA, et al. Do pharmacist-led medication reviews in hospitals help reduce hospital readmissions? A systematic review and meta-analysis. Br J Clin Pharmacol 2016;82:1660–73.doi:10.1111/bcp.13085 Cited

Joseph T, Hale GM, Eltaki SM, et al. Integration Strategies of Pharmacists in Primary Care-Based Accountable Care Organizations: A Report from the Accountable Care Organization Research Network, Services, and Education. Journal Of Managed Care & Specialty Pharmacy. 2017;23(5):541-548. doi:10.18553/jmcp.2017.23.5.541.

While this publication had lots of good information regarding ACOs, we opted not to cite it as our submission was not limited to primary care practices in an ACO.

Shelley Otsuka, Jennifer N. Smith, Laura Pontiggia, Radha V. Patel, Susan C. Day & David T. Grande (2019) Impact of an interprofessional transition of care service on 30-day hospital reutilizations, Journal of Interprofessional Care, 33:1, 32-37, DOI: 10.1080/13561820.2018.1513466 Cited.